# Identification of Health Needs in Ukrainian Refugees Seen in a Primary Care Facility in Tenerife, Spain

**DOI:** 10.3390/nursrep15010027

**Published:** 2025-01-17

**Authors:** Willian-Jesús Martín-Dorta, Cristo-Manuel Marrero-González, Eva-Lourdes Díaz-Hernández, Pedro-Ruymán Brito-Brito, Domingo-Ángel Fernández-Gutiérrez, Oxana-Migalievna Rebryk-De Colichón, Ana-Isabel Martín-García, Estrella Pavés-Lorenzo, María-Candelaria Rodríguez-Santos, Juan-Francisco García-Cabrera, Janet Núnez-Marrero, Alfonso-Miguel García-Hernández

**Affiliations:** 1Primary Care Management Board of Tenerife, The Canary Islands Health Service, 38204 Santa Cruz de Tenerife, Spain; domingofernandez.gaptf@gmail.com (D.-Á.F.-G.); oxanarebryk@hotmail.com (O.-M.R.-D.C.); anaisabel.mar.gar@gmail.com (A.-I.M.-G.); enferestre@hotmail.com (E.P.-L.); m.c.rguez.santos74@gmail.com (M.-C.R.-S.); garcia_cabrera3@hotmail.com (J.-F.G.-C.); janetnunmar@gmail.com (J.N.-M.); 2Nursing Department, University of La Laguna, 38200 Santa Cruz de Tenerife, Spain; pbritobr@ull.edu.es (P.-R.B.-B.); almigar@ull.edu.es (A.-M.G.-H.); 3Nuestra Señora de Candelaria School of Nursing, The Canary Islands Health Service, 38010 Santa Cruz de Tenerife, Spain; eludias@gmail.com; 4Training and Research in Care, Primary Care Management Board of Tenerife, The Canary Islands Health Service, 38204 Santa Cruz de Tenerife, Spain

**Keywords:** migrants, refugees, health services need and demands, primary health care, humanitarian crisis

## Abstract

Background: Ukrainian refugees fleeing the conflict between Russia and Ukraine may face significant challenges to their physical, psycho-emotional, social, and spiritual wellbeing. Aim: To identify the health needs of Ukrainian refugees seen in primary care facilities in Tenerife, Canary Islands, Spain. Methods: A mixed-methods design was employed. Quantitative data were obtained through a descriptive analysis of health records, while qualitative data were collected via focus group interviews and thematic analysis of testimonies. Results: The sample comprised 59 individuals (45.4% of all patients seen). Eight participants from five family groups took part in the focus group. The typical profile of a Ukrainian refugee in the Canary Islands is female (79.7%), relatively young, with a high socio-cultural background, generally in good health, travelling alone or with her minor children. The main reasons for consultation were routine health check-ups and control blood tests. The NANDA-I nursing diagnoses indicated a need for psycho-emotional care, with the most prevalent being Risk for Relocation Stress Syndrome (27.1%); Interrupted Family Processes, Disturbed sleep pattern, Risk for Impaired Resilience (13.6% each); and Anxiety (11.9%). Participants rated the healthcare system positively, but language barriers and long waiting times for access to specific services were noted as limitations. The primary social demands include seeking employment, learning the language, and increasing support groups among Ukrainians themselves. Conclusions: This study underscores the need for a tailored approach to refugee care, considering their unique circumstances and needs. Early provision of information about available healthcare services and protocols can facilitate access, manage expectations, and aid decision-making.

## 1. Introduction

The military operations carried out during the armed conflict between Russia and Ukraine have forced millions to flee their homes, seeking asylum as refugees in other welcoming nations, including Spain. By June 2024, it was estimated that approximately 6.5 million individuals had left Ukraine [1]. This situation has resulted in a rapidly escalating humanitarian crisis. Against this backdrop, the European Union has decided to activate the Temporary Protection Directive in response to a mass influx of displaced persons. This emergency measure offers immediate and collective protection to those unable to return to their country of origin, ensuring the right to residency, access to the labour market and housing, access to education for children, and the right to healthcare services [2]. Additionally, Council Implementing Decision (EU) 2022/382 of 4 March 2022 [3] grants the right to temporary protection to (a) Ukrainian nationals who were residing in Ukraine before 24 February 2022; (b) Stateless persons and nationals from third countries, other than Ukraine, who were beneficiaries of international protection or equivalent national protection in Ukraine prior to 24 February 2022; and (c) Family members of the persons mentioned in points (a) and (b).

Armed conflicts have devastating and multifaceted effects, impacting social life, the environment, and the health systems and infrastructure of affected countries. Such conflicts result in significant short-, medium-, and long-term consequences for victims, with a considerable impact on their physical, psychological, emotional, social, and spiritual health and wellbeing [4,5]. Those forcibly displaced are often required to rebuild their lives in cultures different from their own, a process that is generally slow and painful [6]. According to the Global Action Plan 2019–2023 for promoting the health of refugees and migrants—developed by the World Health Organisation (WHO) in collaboration with the International Organisation for Migration, the United Nations High Commissioner for Refugees (UNHCR), and other international organisations—refugees possess the same universal human rights and fundamental freedoms as everybody else but face specific challenges and vulnerabilities. To address these needs, it is essential to ensure universal health coverage and the highest attainable standard of health [7]. Humanitarian crises disrupt healthcare provision for the affected population, necessitating efforts by host countries to provide inclusive and comprehensive care for the health and wellbeing of refugees and migrants. The circumstances and public health challenges faced by refugees are unique to these populations and are influenced by the different stages of the migration and displacement cycle [8]. In the specific context of forced displacement due to armed conflict, physical pathologies are compounded by psycho-emotional conditions arising from factors such as harassment, traumatic experiences, the ordeal of fleeing, and the very status of being a refugee. This leads to a depletion of the psychological and emotional resources of the refugee population, undermining their natural resilience to illness [9,10].

Previous studies assessing the healthcare needs of the Ukrainian refugee population indicate that the most prevalent needs include vaccination and medication coverage, psychological support, care for respiratory infections, injuries and trauma, access to specific treatments such as chemotherapy or dialysis, management of comorbidities and chronic conditions, and even support for women who are victims of sexual violence or trafficking [11,12,13]. Stiegler et al. [14] also point out that the war in Ukraine represents a polytraumatic event that particularly affects the psychological-emotional sphere. Regarding mental health issues within this population, various studies have reported high rates of obsessive-compulsive and depressive symptoms, post-traumatic stress, disturbed sleep, and elevated levels of distress, especially when there are challenges in integrating into the host country [15,16]. This increased vulnerability to mental health problems makes it a priority for host countries to provide psychological support to the refugee population, particularly given that the Ukrainian healthcare system sustained significant damage to its mental health services in the early stages of the conflict [17,18,19].

Over the past 15 years, the Canary Islands (Spain) have managed multiple crises stemming from migratory movements, primarily involving people arriving from the coasts of the African continent. In 2023, the islands witnessed unprecedented arrival figures, with around 15,729 immigrants arriving in just one month [20]. Currently, there is an intensification of these irregular maritime migrations, which use the Canary Islands as a gateway to access mainland Europe. This poses a significant challenge in terms of planning and managing a phenomenon that is now considered structural within this Spanish autonomous community [21,22]. Despite the presence of common care and healthcare needs across most populations displaced as refugees to other countries [12,13], each migratory phenomenon possesses its own unique characteristics, necessitating tailored and specific care procedures for each group [23]. In March 2022, with the arrival of the first Ukrainian refugees to the Canary Islands, the need arose to understand the scale of this migratory phenomenon and to identify the healthcare needs and care demands of this population.

Spain, like other countries, has developed guidelines for the reception of refugees from Ukraine, which include recommendations based on the vulnerability criteria of migrant populations. These guidelines prioritise the need for initial reception to provide urgent care, assess priority healthcare needs, ensure the availability of regular medications, provide information about the host country’s healthcare system, control the risk of infections and the spread of communicable diseases, offer psychological care, and ensure vaccination coverage [24,25]. A relevant aspect that may influence equal access to and use of healthcare services by refugees is the level of health literacy (HL). HL has been identified as a social determinant of health that affects access to healthcare systems on equal terms, determines autonomy in decision, facilitates understanding of health information, and enhances the safety and quality of self-care [26,27]. In Spain, some initiatives have been launched to promote HL aimed among refugees. These include workshops on the functioning of the healthcare system, healthy habits and disease management [28]. However, in Canary Islands, HL programs have not been systematically implemented for this population.

For both adult and child migrant refugees, primary care services and emergency services generally represent their first point of contact with the Spanish healthcare system [12]. In this setting, nurses, particularly family and community health nurses, are at the forefront as providers of care [29,30,31]. Their training equips them to engage professionally in the shared care of the health of individuals, families, and communities from all life stages, encompassing various aspects of health promotion, disease prevention, recovery, and rehabilitation. Their key roles include the delivery of public and community healthcare, epidemiological and environmental surveillance, and responding to emergency and disaster situations [32,33]. When approached from this community perspective and in the specific context of an influx of refugees due to armed conflict, the effectiveness of nursing care requires the proper identification of the most prevalent issues to effectively plan care and anticipate the potential social and health needs associated with this phenomenon.

The objectives of this study are (1) to determine the healthcare needs of the Ukrainian refugee population attending primary care consultations in a health district within the area of Tenerife (Canary Islands, Spain); (2) to identify the most prevalent NANDA-I nursing diagnoses; (3) to analyse the main barriers to accessing the healthcare system; and (4) to describe the profile of refugees from the armed conflict in Ukraine who attend primary care consultations, based on sociodemographic and clinical variables.

## 2. Materials and Methods

### 2.1. Design

A mixed-methods study was conducted, combining a quantitative approach—featuring descriptive analysis through data collection and extraction from electronic health records (EHRs)—with a qualitative approach, utilising focus group interviews and testimonial analysis. Thus, a sequential exploratory methods approach is employed, beginning with the collection of quantitative data followed by a qualitative research phase, and subsequently integrating or linking the data from the two separate datasets [34]. This manuscript was prepared in accordance with the COREQ guidelines for reporting qualitative research and the Consensus Reporting Items for Studies in Primary Care (The CRISP Statement) [35].

### 2.2. Setting and Participants

The study was carried out at the Casco-Botánico primary care facility located in the northern area of Tenerife, i.e., Puerto de la Cruz (Canary Islands, Spain). The study commenced on 1 March 2022 and concluded on 29 January 2024. Puerto de la Cruz is a tourist municipality in the province of Santa Cruz de Tenerife, part of the Canary Islands autonomous community. In January 2023, the population of this municipality was 30,849 inhabitants [36]. The Casco-Botánico primary care facility is one of 42 basic healthcare districts in Puerto de la Cruz that comprise the primary healthcare map of the island of Tenerife. Since the arrival of the first Ukrainian refugees to the island, a specific service was established to cater to this population, staffed by a Ukrainian-born physician and two nurses. Active dissemination of information about the launch of this service was carried out among non-governmental organisations (NGOs) and the users of the primary care facility. The reception of refugees arriving in the municipality and assistance with processing temporary protection documentation was managed by the Red Cross organisation, which regularly informed the primary care management board and the primary care facility itself of the number of individuals arriving in Puerto de la Cruz. As of June 2022, the number of Ukrainian refugees registered with the Casco-Botánico primary care facility was 478, including minors.

The sample used to assess the sociodemographic and clinical profile of this population comprised adults aged 16 years and older who had been displaced due to the armed conflict in Ukraine and sought adult care at the Casco-Botánico primary care facility. All participants were required to meet the condition of refugee status in Puerto de La Cruz, Tenerife (Canary Islands, Spain). The sample was selected through purposive, non-probabilistic sampling based on the following inclusion and exclusion criteria:

Inclusion criteria:–Individuals aged 16 years or older.–Fulfilment of the temporary protection criteria as outlined by the Spanish Order PCM/170/2022 of 9 March, publishing the Agreement of the Council of Ministers of 8 March 2022, which extends temporary protection under Council Implementing Decision (EU) 2022/382 of 4 March 2022 to those affected by the conflict in Ukraine seeking refuge in Spain.–Availability of an interpreter to ensure effective communication if the participant could not speak Spanish.

Exclusion criteria:–Refusal to provide informed consent.–Individuals with cognitive disabilities who were unable to understand the information sheet or give informed consent, and who were not accompanied by a legal guardian or representative.–Individuals relocated to other areas of residence, outside the Puerto de La Cruz basic health district.

For the narrative exploration phase, which involved discourse analysis through a focus group session, a purposive, non-probabilistic sample of eight participants from five family units was selected, all of whom met the inclusion criteria and agreed to take part in this phase of the study. The number of participants was determined following the recommendations found in the literature on focus group design [37,38,39]. Participants of different age groups, sexes, and familial circumstances were selected to ensure a broad and representative overview of the target population. The group interview was conducted on 21 September 2023 at the Puerto de La Cruz Red Cross facility.

All participants who met the inclusion criteria and none of the exclusion criteria were asked to provide their informed consent in writing, which was translated by a sworn translator as required by the relevant Ethics Committee. If the participant’s ability to understand the study objectives and the basic information provided was deemed insufficient to ensure informed consent, or if they were unable to verbally express agreement or ask questions regarding the terms, the informed consent form was signed by a legal guardian or authorised representative, in accordance with Organic Law 3/2018 of 5 December on the Protection of Personal Data and Guarantee of Digital Rights, and the provisions of Regulation (EU) 2016/679 of the European Parliament and the Council of 27 April 2016 on Data Protection (GDPR).

### 2.3. Variables

To describe the participants’ sociodemographic profile, the following variables were explored: sex, age, marital status, number and relationship of family members who remain in Ukraine (up to second-degree relatives), number and relationship of family members who accompanied the participant to the host location (up to second-degree relatives), level of education, and current employment status. Additionally, the reasons for seeking primary care were recorded.

To identify their health issues and care needs, records from their EHRs were used. These were documented by general practitioners following the International Classification of Diseases, 11th Revision (ICD-11), and by nursing staff according to the NANDA-I nursing diagnoses classification [40].

Furthermore, through a focus group interview, health conditions related to the displacement caused by the armed conflict were explored, along with the opinions and expectations of Ukrainian refugees regarding the healthcare services provided. Our aim was to identify the main barriers to accessing healthcare services and the support networks available to them on the island of Tenerife. Appendix A contains the questions used to guide the focus group interview.

### 2.4. Data Collection

The variables used to describe the sociodemographic profile, identify health issues, and record nursing diagnoses were collected during primary care consultations for Ukrainian refugees. Prior to the consultations, the physician explained the study objectives and key details to them and requested their informed consent. Both the process of obtaining their informed consent and the primary care consultation itself were conducted in the participant’s language. The nursing assessment was carried out using simultaneous interpreting by the physician. At the time of the consultation, participants did not have prior EHRs in the local system. This meant that current clinical and personal data were primarily obtained based on the participants’ self-reports. While self-reporting was essential for identifying current or potential health issues by nursing staff, most diagnoses were made following a systematic nursing assessment based on Marjory Gordon’s health patterns framework [41].

To facilitate access to these consultations, information was actively disseminated through NGOs operating in Puerto de La Cruz, primarily the Red Cross Tenerife, and shared via Ukrainian patients themselves. A field notebook was developed to assist healthcare professionals in recording both EHR-derived data and additional information not documented in the system.

To ensure participant anonymity, the number of variables that could potentially identify individuals was minimised, and access to confidential information was restricted to the research team directly involved in the study. Each participant was assigned a unique identification code to access their health records, as data collection occurred during routine primary care consultations. This code was linked to an alphanumeric identifier, which was used to associate clinical and sociodemographic data relevant to the study, without being tied to any personal identifying information. The coding system was securely maintained by the principal investigator. Data access was strictly controlled, with each researcher only having access to the information necessary for their specific role, ensuring that any details that could lead to re-identification remained inaccessible to those not requiring them.

For the focus group interview, the research team collaborated with the Red Cross Tenerife, which provided social support to Ukrainian refugees in Puerto de La Cruz. The study objectives were presented to the organisation, and their help was requested to recruit participants based on the study’s target profile. A formal agreement was obtained from the organisation to facilitate this phase of the study. On 22 May 2023, a meeting was held at the Red Cross headquarters in Puerto de La Cruz, where potential candidates were invited to participate. During this meeting, the lead investigators explained the study objectives and addressed any methodological and ethical concerns. When necessary, simultaneous interpreting was provided by the study’s physician and co-investigator. Following the meeting, a list of potential participants from various family units who expressed willingness to participate was compiled. From this list, the research team selected eight participants, representing five family units, to participate in the focus group session. The selected participants were contacted again to review the study’s objectives, methodology, and ethical aspects. Their informed consent and permission for recording the session were also obtained using a sworn translation of the form. The research team ensured confidentiality and participant anonymity, with data handled in compliance with legal regulations.

As previously mentioned, the focus group interview took place on 21 September 2023 at the Red Cross headquarters in Puerto de La Cruz, in a safe and already familiar environment for participants. A semi-structured interview guide (Appendix A) was used, allowing participants to share their experiences and perceptions openly, while the researchers focused on specific topics of interest. The session was recorded, transcribed, and analysed using thematic coding to identify patterns and trends in the responses. These codes and categories were documented in the codebook and hierarchical map. Simultaneous interpreting was provided throughout the session by the physician.

Five researchers participated in this phase. Researcher 1 was the main moderator, leading the interview and posing the established questions. Researchers 2 and 3 were responsible for coordinating spealing turns, manging the recording, and assisting Researcher 1 as the primary moderator. Researcher 4 conducted the simultaneous translation of the narratives. Researcher 5 oversaw logistical support during the interview. Researchers 1, 2 and 3 served as the primary observers. The seating arrangement of participants and investigators in the interview room is shown in Figure 1.

### 2.5. Data Analysis

Data on the sociodemographic and clinical variables collected during the primary care consultations were stored in a database created using the statistical software SPSS^®^ v.29.0, where they were subsequently cleaned and processed. For the descriptive phase, nominal variables were analysed using frequency distributions, while numerical variables were summarised using means (SDs) or medians (ranges), depending on their distribution. To explore differences in consultation reasons by sex, Pearson’s chi-square test was applied. All statistical tests were conducted with a statistical significance threshold of alpha < 0.050.

For the analysis of the textual data obtained from the focus group interview, a peer review was conducted by the two most experienced researchers on the team in qualitative analysis. Each researcher independently reviewed the verbatim transcripts and performed inductive coding and data categorization, subsequently representing the results in a hierarchical map. This process required multiple reviews of the text to reorganize the information into broader or more specific categories. The qualitative analysis software Nvivo14^®^, version 14 for Windows, was employed to support this process. Based on these categories, subcategories, and their interrelations, each researcher developed a conceptual structure. The results of this process were then shared and cross-checked with the narratives to carry out, through consensus, the analysis and interpretation of the information.

## 3. Results

### 3.1. Sample Description

Between 12 March and 30 June 2022, a total of 243 appointments were recorded, including both scheduled and walk-in consultations. After excluding 18 appointments due to duplication or incorrect registration, a total of 225 Ukrainian patients (76.5% of whom were women) were registered for primary care consultations at the Casco-Botánico primary care facility. Of these, 130 patients attended their appointment, while 95 did not (42.2%).

The final sample used for analysis consisted of 59 participants—47 women (79.7%) and 12 men (20.3%)—who met all the inclusion criteria, none of the exclusion criteria, and agreed to participate in the study. These represented 45.4% of the 130 patients who attended the consultation. Surprisingly, the most common reason for exclusion was refusal to sign the informed consent form, despite having received all necessary information in their native language. This occurred in 60 participants (46.2%). The remaining exclusions were due to other established criteria.

The mean age was 46.9 (15.9) years; 47.1 (15.0) years for women and 45.8 (20.1) years for men. Regarding marital status, 61.9% were married or had a partner, 16.9% were single, 6.8% were widowed, 5.1% were separated or divorced, and 10.6% did not wish to provide this information. In terms of level of education, 42.4% had a university degree, 11.9% had technical or vocational training, 10.2% had completed secondary education, 8.5% had primary education, and 27.1% did not provide this information. At the time of the consultation, 59.3% of participants were unemployed, 13% were retired, 3.4% were employed, 1.7% were students, and 13.6% did not provide information regarding their employment status. Table 1 summarises the data on the number of family members (up to second-degree relatives) who remained in Ukraine or had relocated to Tenerife with the participant.

The reasons for which participants sought medical assistance at the primary care facility are summarised in Table 2, with no statistically significant differences by sex for any of the reasons. No participants attended the consultation to request vaccination. The median number of reasons for consultation was 2 (1–5), with no significant differences by sex (*p* = 0.070).

A total of 59 health issues were documented on the EHR, classified according to the International Classification of Diseases (ICD-11) (Appendix B). The average number of health issues per participant was 1.6 (1.6). The most frequently recorded conditions were hypertension and tobacco abuse, both present in 13.6% of the sample. Health problems related to the psychosocial sphere, which may be linked to their forced displacement due to the armed conflict, included anxiety (5.1%); adjustment disorder (3.4%); insomnia (3.4%); panic attacks (1.7%); and depression (1.7%).

A total of 23 NANDA-I nursing diagnoses (Appendix B) were recorded in the EHR. The most frequently documented nursing diagnosis was Readiness for Enhanced Health Self-Management, identified in 42.4% of the sample. NANDA-I nursing diagnoses from the psychosocial domain that were recorded by the nurses, and which may be related to the situation of forced displacement, included Risk for Relocation Stress Syndrome (27.1%); Interrupted Family Processes (13.6%); Disturbed Sleep Pattern (13.6%); Risk for Impaired Resilience (13.6%); Anxiety (11.9%); Insomnia (3.4%); Relocation Stress Syndrome (3.4%); Fear (3.4%); Compromised Family Coping (1.7%); Ineffective Health Self-Management (1.7%); Sleep Deprivation (1.7%); Fatigue (1.7%); Ineffective Family Health Self-Management (1.7%); and Risk for Loneliness (1.7%).

### 3.2. Focus Group Results

#### 3.2.1. Genogram and Family Context

The focus group consisted of eight participants (six women and two men) from five family units, with a median age of 41.5 (34–72) years. Family units 1, 2, and 4 were represented by two participants each, while units 3 and 5 were represented by a single participant. All family units, up to second-degree relatives, were incomplete, primarily due to the absence of young men who remained in Ukraine participating in the conflict. Figure 2 shows the genogram for each family member by family unit, all of whom were refugees in the municipality of Puerto de La Cruz, Tenerife [42].

All participants reported having lost family members, friends, neighbours, or acquaintances as a result of the war, and mentioned that they were constantly awaiting updates from Ukraine regarding the status of their loved ones and the progress of the conflict.

#### 3.2.2. Discourse Analysis

Figure 3 shows the hierarchical map representing the categories derived from the discourse analysis of the focus group interview, illustrating the weight of participants’ responses. Appendix C includes the codebook with the labels assigned to data segments representing key ideas, concepts, or themes, along with the number of references categorised under each label.

#### Social Integration and Community Support

The focus group results suggest that Ukrainian refugees share similar needs, assessments, criticisms, and suggestions. Their reception in Tenerife, managed by the Red Cross and the Spanish Commission for Refugee Assistance (CEAR), ensured that from their arrival, they were informed about their entitlements and received immediate aid. Initially, most refugees were housed in hotels, later relocating to accommodation provided by public and private institutions. Information about social and economic aid primarily came through NGOs and a shared instant messaging group among refugee families in the Canary Islands.

The refugees expressed positive views on the support received from the local population, describing them as kind and welcoming. As Participant 8 noted: ‘[…] people are very open, very kind to us […] they understand our situation and always want to help’. However, participants highlighted a lack of mutual support networks among the refugees, attributing this to mistrust and concern for the safety of family members still in Ukraine. This mistrust, they said, limits the development of peer support communities, which they see as essential for social and emotional well-being. Participant 4 remarked: ‘[...] we receive more support from the Spanish than from among ourselves’.

#### Healthcare

As shown in the hierarchical map (Figure 2), a descriptive analysis of the dimensions identified through coding reveals that Ukrainian refugees evaluate healthcare in the Canary Islands primarily based on quality and the availability of resources. Participants expressed a desire to be included in health promotion and prevention programmes without having to request them themselves. They also called for faster and more streamlined access to diagnostic tests. Participant 5 stated: ‘[doctors here] don’t offer [these things], they need to offer [them]. If you need something, you have to ask for it; otherwise, the doctors assume it’s not necessary […] why doesn’t the doctor or paediatrician suggest it? We’ve been here for a year, so why hasn’t anyone suggested a blood or urine test for the child […]?’

#### Access to Medication and Services

Participants explained that, in Ukraine, they used both public and private healthcare services to reduce waiting times and gain access to more complex diagnostic tests. Upon arriving in the Canary Islands, they generally encountered no difficulties accessing primary care consultations or having their regular medications prescribed, except in cases where medication funding required special authorisation or approval. However, some participants mentioned turning to private healthcare due to delays in accessing certain specialists and the lack of diagnostic tests being offered. Participant 6 shared: ‘Yes, they’ve given her [my mother] her medication, but there’s one medication that the nurse or doctor said isn’t available here, so we’re waiting [...] and they didn’t offer an alternative, and her appointment with neurology or psychiatry isn’t until October.’ Participant 1 noted: ‘[...] gynaecology appointments are harder to get than in Ukraine [...] you have to wait a few years or wait three months for a phone consultation, it’s harder. In Ukraine, you can call and have the problem solved within a week or two, because there are urgent cases that need [immediate] attention.’ Participant 8 added: ‘For us, it’s a bit strange to [have to] wait six months, a year, or more for an appointment, especially for older people, like [our] parents or the elderly, they can’t imagine having to wait a year or more. But we have to live and keep up with the pace in Spain because we live here now; in our situation, we can’t choose, we just have to keep going, I think.’

#### Perception of Healthcare Professionals

Participants highlighted and appreciated the kindness, approachability, and good treatment received from healthcare professionals. They also emphasised the professionals’ willingness to communicate in other languages. Participant 8 stated: ‘[…] we are foreigners, not everyone speaks enough Spanish, and sometimes the doctors don’t speak English, but they try to learn and help [...].’

#### Perception of Physical and Emotional Wellbeing

Most participants described their physical health as good, with only older individuals rating it as fair or poor. Regarding emotional well-being, the majority reported higher levels of distress, although they did not specify particular symptoms. They expressed that their emotional state largely depended on the news they received from Ukraine. Participants 3 and 8, respectively, noted: ‘Emotionally, [we have] ups and downs depending on the news we receive’ and ‘My father gets very nervous when hearing news from Ukraine.’

#### Mental Health

The demand for professional psychological care from the group was not significant. Disturbed sleep and rest, nightmares, headaches, eye pain, and hypertensive crises were mentioned as health issues they associate with the impact of the armed conflict. Participant 5 stated: ‘[…] yes, they [my parents] have nightmares about the war because they lived through it... because they come from a city where they experienced it.’

Some participants believe that mental health issues should only be addressed by psychologists and psychiatrists, not by primary care physicians or nurses. They see overcoming the language barrier as crucial for mental health care, citing it as their main obstacle. If healthcare professionals do not speak English or communication in Spanish is not possible, they prefer not to use an interpreter when discussing mental health issues. One participant mentioned trying to use an online machine translator to communicate with the physician to avoid sharing their experience with a third person, but the physician rejected this method. They generally expressed reluctance to discuss the psychological effects of the war, not only with professionals but also with other Ukrainian refugees. They noted that primary care professionals usually do not inquire about their emotional state, but they do not expect them to. Participants 3 and 4 remarked: ‘[…] that’s the psychologist’s job, each specialist does their own thing, and of course, when it comes to talking to a psychologist, we don’t feel confident enough to do it in another language.’ They added that, upon arriving on the island, the Red Cross provided psychological care, but the professionals only spoke Spanish, so they did not use the service.

#### Perception of the Care Received in Emergency Services

Participants expressed very positive views regarding the care received in emergency services, both at the primary care level and in hospitals when referred. They mentioned using these services not only to address acute health issues but also to try to expedite treatment for chronic conditions. They placed significant value on the immediacy of urgent care, as noted by Participant 1: ‘Emergency services work the best [...] they’re quick.’

#### Main Demands

The main demands expressed by the participants regarding healthcare relate to the waiting times for accessing certain medical specialties, both in community settings and hospitals, as well as specific services offered in primary care, such as midwifery and dentistry. Participants also believe that primary care professionals should more frequently offer diagnostic and screening tests, with a particular emphasis on laboratory tests as part of routine health check-ups. The demand for health education and psychological or emotional care was not significant among the participants. They do not consider mental health care to be the responsibility of primary care physicians or nurses, and they emphasised the importance of ensuring effective communication without intermediaries, in order to maintain confidentiality. The language barrier was identified as a key challenge for Ukrainian patients in accessing healthcare services; however, participants appreciated the willingness and effort of healthcare professionals to facilitate communication as effectively as possible. The immediacy of care was highly valued and often compared with the healthcare services in Ukraine. As Participant 8 remarked: ‘The only difference, if we compare here with Ukraine, is that in Ukraine it’s a bit quicker. We can get appointments in a week or three days. We don’t have to wait [that long]. And with private healthcare, you call and see the doctor the next day.’

Regarding social demands, the need to find work and learn the language were the most prominent. Participants considered the Canary Island population to be socially friendly and welcoming, and they highly valued the support they received. However, they noted a significant lack of support groups among the Ukrainian community itself, expressing concerns and a lack of trust in sharing personal information or updates about their families with other refugees on the island.

## 4. Discussion

The findings of this study, along with those from other similar studies, highlight that refugee populations are diverse, and so are their healthcare needs. These differences depend on various factors, including the sociodemographic characteristics of the population seeking asylum, the nature of the event causing displacement, the reception and protection measures provided by host countries, cultural proximity, and the social perception of the migration phenomenon itself [24,43,44,45,46,47]. The recent increase in studies aimed at assessing the healthcare needs of refugee populations [43,44,45,46] reflects growing concern about understanding the specific characteristics of each phenomenon to ensure appropriate access to health services, without neglecting the priority given to other vulnerable populations. It is also essential not to limit the analysis to strictly healthcare-related problems, but to consider the social determinants impacting health, such as access to financial aid, secure housing, employment, or education [46,47].

The results of this study show that the profile of Ukrainian refugees in the Canary Islands is predominantly female, of non-advanced age, often travelling alone or with one or two family members, frequently children, with adult males typically remaining in Ukraine. The educational level of the sample is notable, with over 60% having higher or secondary education. This profile is similar to that reported in other studies of Ukrainian refugees in other European countries [15,45,46].

The average of 1.6 health issues per patient in our study may be related to the predominantly younger age group, where a lower frequency of comorbidities is expected. Furthermore, the high educational level may be associated with greater financial resources, enabling better healthcare coverage in their home country. According to the WHO, although basic healthcare is free in Ukraine, around 50% of care in 2018 was provided by the private sector [48]. Evidence suggests that the socioeconomic and cultural profile of Ukrainian refugees differs from internally displaced persons, with those leaving the country having higher social and economic status, greater cultural capital, and more connections abroad [49]. A study examining the financial and social determinants of health in highly educated Ukrainian women who are refugees in the Czech Republic reported that around 55% of them rated their health as good or very good, with 77% reporting between zero and two illnesses [46]. Predictors of poor health included the number of health issues, the ability to receive support from others, and the severity of depressive symptoms. Although our study included male participants (20.6%), the comorbidity rate was similar, with 76.3% of participants having fewer than two health conditions.

Studies identifying the reasons for consultation or health problems in the Ukrainian refugee population, beyond those related to mental health, remain limited. A study in Germany found that primary care physicians attending Ukrainian refugees reported the most common reasons for visits were requests for regular medications (80.8%), diagnosis and treatment of acute health issues (69.2%), completion of vaccinations (40.4%), and requests for preventive tests (21.2%) [50]. Similar findings were reported in Poland, where the predominant reasons for consultations were renewing regular treatments and addressing cardiovascular diseases and diabetes complications [51]. These studies linked the high demand for regular treatments to the significant burden of non-communicable chronic diseases in Ukraine, primarily cardiovascular diseases, diabetes, and mental health issues [50]. Other studies on internally displaced Ukrainians due to the armed conflict also highlight a clear relationship between comorbidity and age, particularly in older adults (over 60 years) [52]. In our study, most participants attended consultations for routine health check-ups (89.8%) and/or control blood tests (52.5%). These high percentages are understandable, given that these two services were the most widely promoted among Ukrainian refugees in Puerto de La Cruz. Requests for regular medications were recorded less frequently (42.4%) than in other studies [50,51]. Additionally, 39.1% of patients with scheduled appointments did not attend. While this no-show rate may be explained by the initial mobility of refugees within the island of Tenerife, it could also indicate a younger and healthier patient profile who decided not to use the healthcare services offered.

Contrary to other studies [50,51,52], the sample in our study presented fewer chronic health conditions, with hypertension (13.5%) and tobacco abuse (13.5%) being the most prevalent. Diabetes was recorded at lower rates than the estimated prevalence in Ukraine, which is estimated to be at 9.1% [53]. Several authors link the significant heterogeneity of the burden of disease among Ukrainian refugees to their age, sex, place of origin in Ukraine, and socioeconomic status. A study estimating the prevalence of diseases among internally displaced persons and refugees in other countries found a 4.1% prevalence of diabetes among Ukrainian refugees in Europe, with higher prevalence in those over 70 years of age [47].

During the evaluation period, no participants attended primary care consultations requesting vaccinations, with a particular reluctance to complete COVID-19 vaccination schedules. In Ukraine, the full vaccination rate (two doses) for COVID-19 at the start of the war was 34%, with a booster vaccination rate of 2% [54]. The WHO points to widespread scepticism towards vaccination among the Ukrainian population, not only against COVID-19 but also for other vaccines, such as those for measles, diphtheria, tetanus, or polio [55], which may indicate a considerable number of pockets of under-immunised population. While the initial documentation of communicable diseases in our sample was not significant, it is important to note that the information recorded in the EHR largely depended on self-reports from the participants. This could be particularly sensitive when declaring communicable diseases such as HIV, hepatitis, or tuberculosis, among others.

The recording of care needs identified by nurses is carried out using the nursing diagnoses of the NANDA-I classification. This classification brings together the best criteria for the diagnosis of care needs, as confirmed by several authors [56,57]. It has been translated into numerous languages and is utilized by thousands of professionals to reflect the care issues underpinning their care work. In the recent decades, it has been incorporated into electronic health records as a standardized language that meets sufficient quality criteria for the documentation of nursing diagnoses in various institutions, health services, and organizations such as the Spanish Ministry of Health [58]. Furthermore, it has been demonstrated that the NANDA-I classification is aligned with other nursing intervention and outcome classification systems, such as NIC and NOC, which reinforces its applicability in clinical practice [59].

The identification of the NANDA-I nursing diagnoses resulting from nursing assessments—in line with medical records—did not initially suggest numerous care needs related to chronic health issues. However, there was a higher number of psychosocial problems, likely attributable to forced displacement and the stress caused by the conflict. Studies evaluating the impact of war and forced displacement on mental health are prevalent in this population [60,61,62,63]. Women and children are among the most vulnerable demographic groups for mental health problems in conflict contexts [64,65], being at high risk of developing a range of disorders, including post-traumatic stress disorder (PTSD), anxiety, distress, depressive symptoms, and disturbed sleep [66,67].

Although there is a lack of studies specifically assessing the care needs of these patients using standardised nursing languages, the NANDA-I diagnoses recorded by the nurses indicate a patient profile with psycho-emotional care needs, consistent with findings from most of the studies reviewed [15,67,68,69,70,71,72]. However, the records made by medical practitioners appear to underestimate the presence of these issues. One of the most notable differences is the frequency of anxiety being recorded: 5.1% by medical practitioners compared to 11.9% by nursing staff. While the nursing data are higher, they are still lower than figures reported in other studies, where anxiety rates in the Ukrainian refugee population exceed 50% [15,67] and even surpass those found in internally displaced Ukrainians [68]. Disordered sleep was also more frequently recorded by nursing staff, at 17.7% compared to 3.4% by medical practitioners, though these figures are still below the 20%+ reported by other authors [67,69,70]. Depression was recorded in the EHR of only one patient by medical staff. Prevalence rates of depression in refugee populations range widely, from 2.3% to 88% [67,70,71,72], depending on sample variability and diagnostic criteria, all of which are higher than the rates found in our study. Other NANDA-I diagnoses recorded by the nurses, including Risk for Relocation Stress Syndrome, Risk for Impaired Resilience, and Interrupted Family Processes, align with the human responses reported in other studies [60,61,62,63,70,73].

In general, both the EHR records and the focus group discussions suggest that the participants primarily sought care for acute issues and disease prevention, with less emphasis on health education or continuity of care for chronic conditions. This focus seems consistent with the patient profile we analysed and the care needs typically prioritised in the early phases of displacement, as highlighted by other studies [70,74]. However, most studies report a higher incidence of mental health-related issues than those recorded in our study. Only one study [74], conducted in a primary care setting, reports similarly low rates (3%) of consultations for psychosocial issues. Some of the reasons cited for these low detection rates include language barriers, time constraints, lack of specific training for healthcare staff to explore and identify these issues, and the absence of systematic assessments. These factors likely apply to our study as well. However, our focus group discussions reveal additional factors worth noting. Firstly, the integration and social support facilitated by the local population in Tenerife were highly appreciated. A sense of security and a good adjustment to the host community’s social system seem to play a significant role in reducing the psychological comorbidities among refugees [61]. Secondly, some participants felt that mental health issues should only be addressed by psychologists or psychiatrists, which may have limited the expression of such issues to primary care professionals. Another aspect to consider is the proportion of men in the sample. Some authors suggest that societal expectations of masculinity in Ukraine encourage men to appear strong and project an image of social desirability, which could have led the male participants to underreport mental health concerns [70]. Additionally, despite the fact that the general practitioner spoke Ukrainian, many patients attended the consultation accompanied by fellow Ukrainians with better language skills or previous experience with the system, which may have hindered the disclosure of sensitive health issues due to a lack of privacy. These same factors could explain the limited mention of mental health problems during the focus group session, as participants also expressed a lack of privacy or trust among their peers during this phase of the study. Based on these insights, effectively addressing mental health issues may require faster access to specialists in the field and strategies that foster patient-professional communication while ensuring privacy and trust [51].

With regard to healthcare, participants expressed positive views on the range of services offered but noted that they were not used to waiting long periods for care. Specifically, they referred to waiting times for referrals to other medical specialties. In Ukraine, before the conflict, access to healthcare was faster, although a significant portion of services were provided by the private sector. Ukrainians with sufficient financial resources could request diagnostic tests on demand and pay to expedite the process. This contrast with the Spanish public healthcare system can lead to a sense of loss of control over managing their own health. The demand for shorter waiting times to access certain healthcare services has been highlighted in similar studies [12,45]. As these authors point out, there is a need for more information about how the host country’s healthcare system functions, the services available in primary care, the existing protocols, and the healthcare resources provided. It is also essential to explain the current state of the host country’s healthcare system to avoid creating false expectations. The participants did not mention waiting times for medical or nursing consultations at the primary care facility, but they did express concerns about waiting times for other services, such as midwifery or dentistry. In this regard, other authors have reported longer waiting times for refugees compared to the regular resident population [43]. However, this difference was not observed in our study. In fact, one of the aims of creating a dedicated service for this population was to prioritise their access to care. Regarding access to regular medication, no significant difficulties were reported, apart from those related to visa processing or the need to replace certain medications with those available in Spain, a situation also noted in other studies [12,45].

The refugees’ assessment of healthcare staff was positive, although they identified the language barrier as one of the main challenges in professional-patient interactions. This issue has been similarly noted by other researchers [45,51]. Nevertheless, participants mentioned the willingness of professionals to communicate using machine translation apps when needed.

Other social needs and demands expressed by participants, such as finding employment, learning Spanish, and increasing support groups among Ukrainians, would benefit from updated evaluation to assess the cultural and social integration of the refugee population on the island after two years of resettlement. Organisations like the Red Cross have provided free resources for learning Spanish and guidance on job-seeking and administrative procedures. Our study’s results indicate good sociocultural adaptation, with some participants even expressing a desire to remain on the island permanently after the conflict in their home country is resolved.

This study presents several limitations that may account for the differences found when compared to similar research. Firstly, there were high rates of incomplete data for the variables used to describe the sample’s sociodemographic profile, with some exceeding 35%. Most of the missing data related to family members who remained in Ukraine or to participants’ education and employment status prior to relocation. Furthermore, despite the interest in understanding the families’ income levels, this information was not collected, as participants showed a moderate reluctance to provide it. Although this limitation has not been reported in other studies, the research team deemed it essential to respect the participants’ privacy and ensure their confidence, avoiding requests for information that could be considered sensitive for any reason.

Secondly, the sample size (59 participants) may not be representative of the entire displaced Ukrainian population in Tenerife. Despite efforts to publicise the healthcare services through a dedicated primary care facility, a significant proportion of participants with scheduled appointments did not attend. Additionally, many patients who did attend the consultation refused to sign the informed consent form, even though the information was provided in their native language. This refusal may be understandable given the specific nature of the conflict. To enhance the robustness of the study, a mixed-methods approach was adopted, including discourse analysis from a focus group interview, which, in our view, strengthens the findings and conclusions. According to other authors, this approach not only allows for the quantification of needs but also enables the exploration of participants’ experiences and perceptions, which enriches the interpretation of data and provides a holistic view of care needs. This approach facilitates a more comprehensive and nuanced understanding of the complexities inherent in healthcare, especially in contexts where patients’ needs are diverse and multifaceted [75,76].

Thirdly, another limitation to consider is the language barrier, which may have affected both data collection and analysis, particularly during consultations and the focus group interview. To mitigate this issue, the collaborating physician/researcher provided interpreting services throughout the study. It is also worth noting that a considerable proportion of the patients already had a sufficient command of Spanish to engage in essential conversations.

Finally, data collection was conducted during the initial phase of the arrival of Ukrainian refugees in the municipality of Puerto de La Cruz (Tenerife, Canary Islands). Many of the findings are related to the need for initial healthcare, and it would be valuable at this point to examine the current healthcare needs and how they are influenced by other social determinants. Nevertheless, the primary focus of this study was on the initial stage of immigration, during which the healthcare system was likely to face greater challenges due to the large influx of refugees.

## 5. Conclusions

This study highlights the importance of adopting a tailored approach to the care of refugees, taking into account their specific contexts and needs. The experience of Ukrainian refugees on the island of Tenerife (Spain) underscores the effectiveness of a well-organised reception system and the crucial role of accessible healthcare services and integration programmes in enhancing their wellbeing. The demands regarding the quality of healthcare services expressed by the participants do not substantially differ from those of the local population. Furthermore, no significant disease burden has been found initially that would strain available resources or hinder access to healthcare services for Ukrainian refugees or other vulnerable groups. Early provision of information about healthcare protocols, services, and available resources, as well as the current state of the local health system, contributes to improved access, prevents unrealistic expectations, and supports informed decision-making.

Addressing the multifactorial health needs of refugee populations requires the proper identification of the most prevalent issues to ensure effective medical care and service delivery. Additionally, understanding the opinions, beliefs, and perspectives of this population regarding the healthcare they receive enables necessary adjustments to be made, ensuring their wellbeing, while fostering social cohesion and integration within the host community.

It is essential to propose new studies that delve into the care needs of the Ukrainian refugee population in host countries at the present time. Considering the duration and characteristics of the ongoing armed conflict, such research should thoroughly explore the transformations experienced by this population, their adaptation to new realities, and the conditions that have either facilitated or hindered their integration. These insights are crucial for designing personalized nursing interventions that effectively respond to the dynamic and evolving needs of this vulnerable population.

## Figures and Tables

**Figure 1 nursrep-15-00027-f001:**
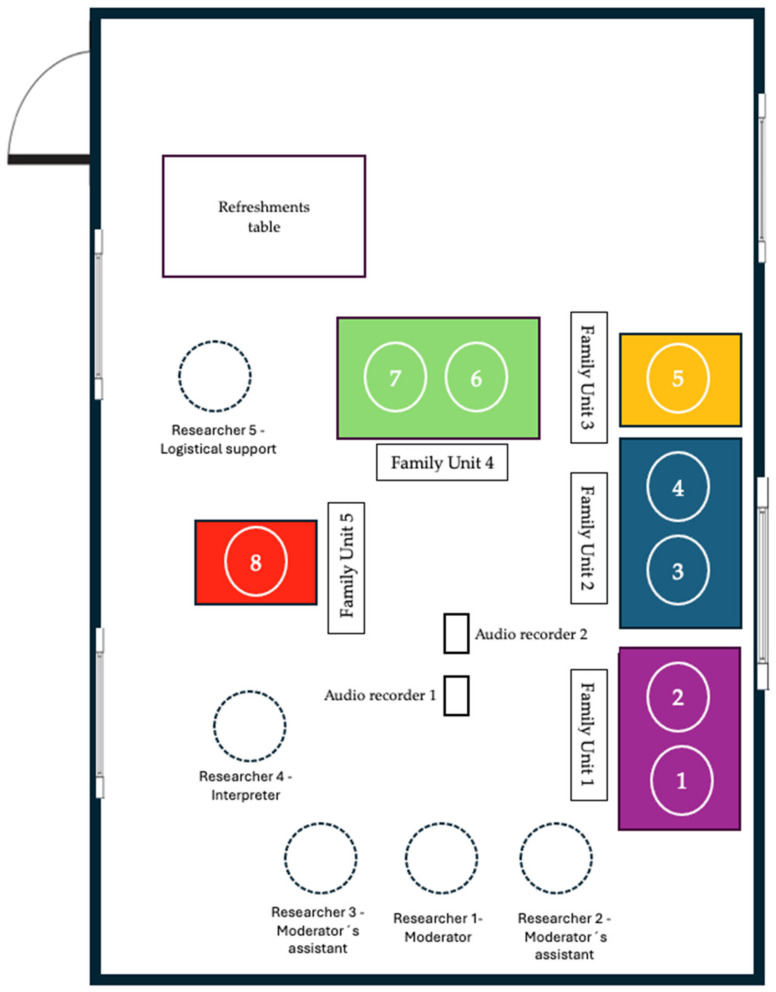
Distribution of researchers and participants by family units in the room during the focus group interview.

**Figure 2 nursrep-15-00027-f002:**
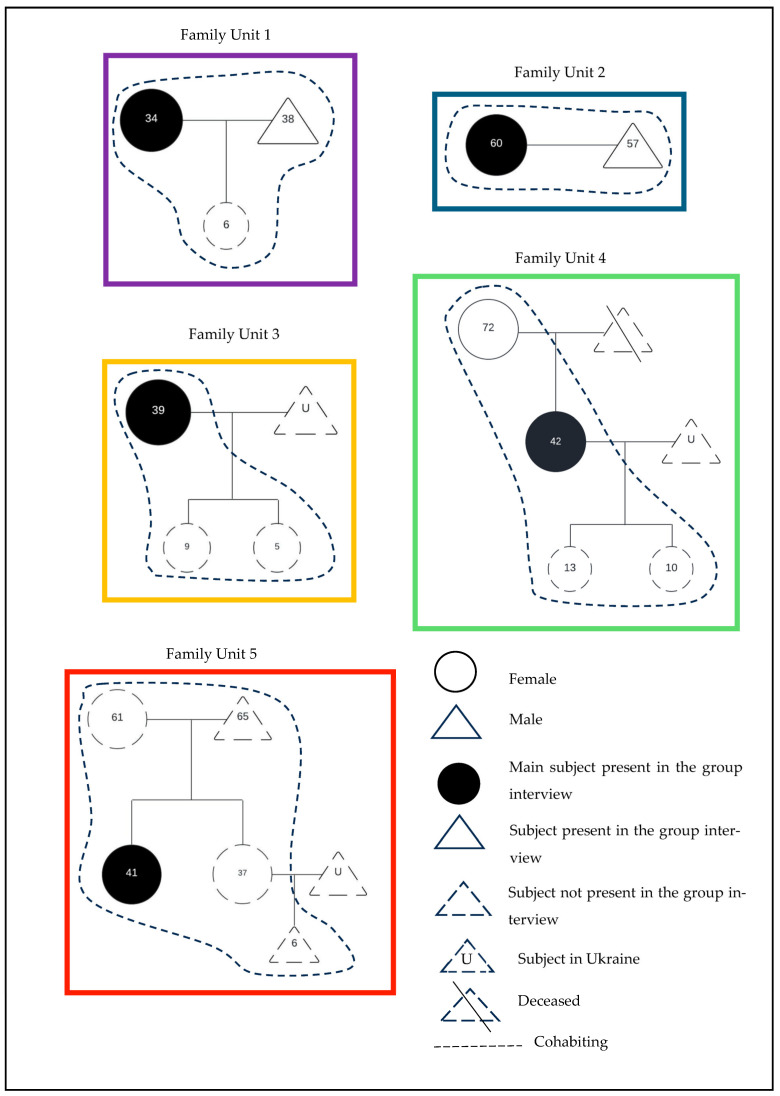
Genogram of the participants.

**Figure 3 nursrep-15-00027-f003:**
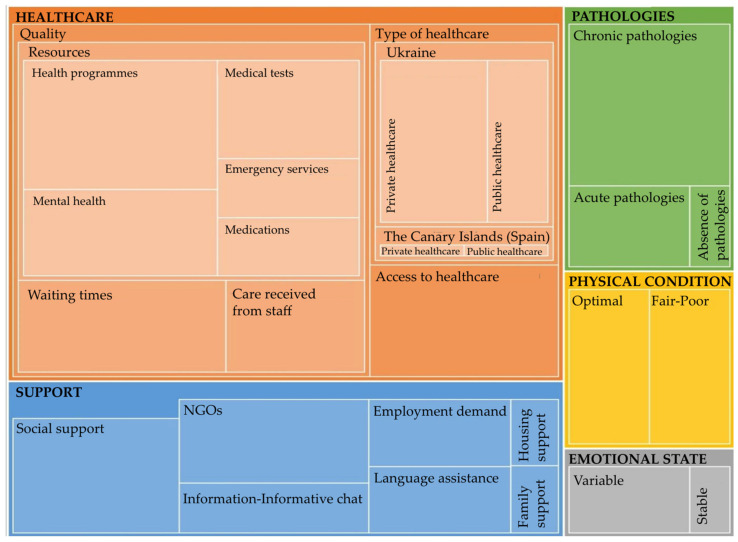
Hierarchical map of discourse categories.

**Table 1 nursrep-15-00027-t001:** Number and relationships of family members (up to second-degree relatives) remaining in Ukraine and accompanying the participant.

Variable	Category	n (%)	Mean (SD)
Number of family members (up to second-degree relatives) remaining in Ukraine	0	3 (5.1)	2.2 (1.3)
	1	7 (11.9)	
	2	15 (25.4)	
	3	7 (11.9)	
	4	4 (6.8)	
	5	2 (3.9)	
	NR	21 (35.6)	
Relationship of family members (up to second-degree relatives) remaining in Ukraine	Partner (male) + son	6 (10.2)	
	Partner (male)	3 (5.1)	
	Partner (male) + son + father + mother	2 (3.4)	
	Partner (male) + son + mother	2 (3.4)	
	Partner (male) + father + mother	2 (3.4)	
	Partner (male) + mother	1 (1.7)	
	Partner (male) + father + mother + sister	1 (1.7)	
	Partner (male) + grandparents	1 (1.7)	
	Father + mother	1 (1.7)	
	Son	1 (1.7)	
	Son + mother	1 (1.7)	
	Son + grandson	1 (1.7)	
	Daughter + granddaughter	1 (1.7)	
	Son + grandsons	1 (1.7)	
	Daughter + sister	1 (1.7)	
	Mother	1 (1.7)	
	Mother + sister	1 (1.7)	
	Father + mother + grandparents	1 (1.7)	
	Father + mother + brother	1 (1.7)	
	Father + mother + brother + grandparents	1 (1.7)	
	Father + brother	1 (1.7)	
	Brother	1 (1.7)	
	NR	22 (37.3)	
Number of family members (up to second-degree relatives) accompanying the participant	0	12 (20.3)	1.5 (1.2)
	1	16 (27.1)	
	2	15 (25.4)	
	3	7 (11.9)	
	4	4 (6.8)	
	NR	5 (8.5)	
Relationship of family members (up to second-degree relatives) accompanying the participant to Tenerife	Partner (male)	5 (8.5)	
	Partner (female)	3 (5.1)	
	Partner (male) + daughter	2 (3.4)	
	Partner (female) + daughter	2 (3.4)	
	Partner (male) + daughter + son	2 (3.4)	
	Partner (male) + mother + daughter	2 (3.4)	
	Partner + children (>2)	1 (1.7)	
	Son	4 (6.8)	
	Children (>2)	1 (1.7)	
	Daughter + sister	1 (1.7)	
	Son + sister	1 (1.7)	
	Daughter + son + sister	4 (6.8)	
	Daughter + grandsons	1 (1.7)	
	Father + mother + sister	5 (8.5)	
	Mother + sister	8 (13.6)	
	Mother	4 (6.8)	
	Sister	2 (3.4)	
	Granddaughter	2 (3.4)	
	NR	1 (1.7)	

NR = Not reported by the participant.

**Table 2 nursrep-15-00027-t002:** Reasons for consultation among participants. Differences by sex.

Reasons for Consultation	Total n (%)	Sex		Pearson’s Chi-Squared	*p*-Value
		Women n (%)	Men n (%)		
General health check-up	53 (89.8)	43 (72.9)	10 (16.9)	0.70	0.40
Request for blood tests	31 (52.5)	25 (42.4)	6 (10.2)	0.04	0.84
Request for regular medication	25 (42.4)	17 (28.8)	8 (13.6)	3.64	0.06
Consultation for an acute health issue and initiation of treatment if necessary	24 (40.7)	21 (35.6)	3 (5.1)	1.53	0.21
Referral for assessment by a specialist other than family medicine	16 (27.1)	13 (22.0)	3 (5.1)	0.03	0.85

## Data Availability

The data presented in this study are available upon request from the corresponding author. The data are not publicly available due to privacy/ethical restrictions.

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
