# Peer review of "Identification of Health Needs in Ukrainian Refugees Seen in a Primary Care Facility in Tenerife, Spain"

_nursrep, 2025, doi:10.3390/nursrep15010027_

Round 1
Reviewer 1 Report
Comments and Suggestions for Authors
Thanks for the opportunity to review this paper. I found it interesting to read, and conceptually sound. The introduction and research aims were clear. The design as a mixed methods study could be enhanced by describing how the methods were mixed and by attending to the use of a reporting guideline such as Improving the Reporting of Primary Care Research: Consensus Reporting Items for Studies in Primary Care-the CRISP Statement | EQUATOR Network.
One other suggestion is to comment a bit more on the role of nurses in documenting NANDA diagnoses. This is not as widely implemented in all healthcare settings, so a bit more detail on this practice, and perhaps an additional limitation might be appropriate.
Author Response
Document with responses to reviewer is submitted

Reviewer 2 Report
Comments and Suggestions for Authors
Kindly read the enclosed file

Author Response
Document with responses to reviewer is submitted.
